# Turning Base Transceiver Stations into Scalable and Controllable DC Microgrids Based on a Smart Sensing Strategy

**DOI:** 10.3390/s21041202

**Published:** 2021-02-09

**Authors:** Miguel Tradacete, Carlos Santos, José A. Jiménez, Fco Javier Rodríguez, Pedro Martín, Enrique Santiso, Miguel Gayo

**Affiliations:** 1Department of Electronics, University of Alcalá, Alcalá de Henares, 28805 Madrid, Spain; miguel.tradacete@edu.uah.es (M.T.); jose.jimenez@uah.es (J.A.J.); pedro.martin@uah.es (P.M.); enrique.santiso@uah.es (E.S.); miguel.gayo@uah.es (M.G.); 2Department of Signal Theory and Communications, University of Alcalá, Alcalá de Henares, 28805 Madrid, Spain; carlos.santos@uah.es

**Keywords:** base transceiver stations (BTS), microgrid, green communications, energy management systems (EMS), IEC61850 standard, embedded systems for Internet of Things (IoT), monitoring and control systems, photovoltaic distributed generation

## Abstract

This paper describes a practical approach to the transformation of Base Transceiver Stations (BTSs) into scalable and controllable DC Microgrids in which an energy management system (EMS) is developed to maximize the economic benefit. The EMS strategy focuses on efficiently managing a Battery Energy Storage System (BESS) along with photovoltaic (PV) energy generation, and non-critical load-shedding. The EMS collects data such as real-time energy consumption and generation, and environmental parameters such as temperature, wind speed and irradiance, using a smart sensing strategy whereby measurements can be recorded and computing can be performed both locally and in the cloud. Within the Spanish electricity market and applying a two-tariff pricing, annual savings per installed battery power of 16.8 euros/kW are achieved. The system has the advantage that it can be applied to both new and existing installations, providing a two-way connection to the electricity grid, PV generation, smart measurement systems and the necessary management software. All these functions are integrated in a flexible and low cost HW/SW architecture. Finally, the whole system is validated through real tests carried out on a pilot plant and under different weather conditions.

## 1. Introduction

In the last two decades, there has been a growing demand for Base Transceiver Stations (BTSs) due to the development of mobile communication networks with smaller cells and BTSs closer to the users. From the network operator (NO) point of view, BTSs are the main source of energy consumption. A decade ago, virtually 60% of the energy consumption of mobile phone operators was directly attributed to the equipment installed in the BTSs [1]. However, with the advent of the fourth (4G) or Long-Term Evolution (LTE) and fifth generation (5G) networks, the amount of traffic volume in the mobile networks has considerably grown, which has led to an increase in the total energy consumption, and therefore, in the carbon footprint generated. Moreover, 5G networks require between two and three times the number of BTSs compared to those installed for the legacy mobile generations [2]. Although 5G has been designed to be more energy efficient than the previous generations [3], the deployment of BTSs for 5G will increase the energy consumption by 5% [4]. As a result, the costs in terms of capital expenditure (CapEx) and operational expenditure (OpEx) will steadily rise. Enhancing the energy efficiency of telecommunication networks becomes a significant contribution when it comes to fighting global warming. However, in the context of rapidly rising energy prices, it is also creating economic opportunities [5]. Currently, energy consumption is regarded as an important performance indicator of the equipment comprising the BTSs [6]. This is stated in the European Telecommunications Standards Institute (ETSI) standard [7] and in 3GPP (3rd Generation Partnership Project) specifications [8].

For many years in the design of mobile networks, the focus has always been on increasing the user throughput and the service provider capacity, without taking energy efficiency and environmental impact into consideration. However, the last decade has seen a change of emphasis and improving energy efficiency within BTSs has become strategically vital, not only for financial reasons, reducing costs, but also for developing a more self-sustainable network of BTSs, which clearly has a positive impact on the corporate brand image of the network operators (NOs). Several projects, such as EARTH [9], ECONET [10], co-funded by the European Commission under the Framework Programme 7 (FP7), and GreenTouch [11] and 5GrEEn [12], have been carried out, which shows the growing public and private concern about this field. Finally, OPERA-Net and OPERA-Net 2 [13] are other projects which stand out in the field of energy efficiency and low-cost implementation for Mobile Radio Access Networks.

In the literature, two approaches have mainly been put forward to address this problem: (i) energy saving strategies, also called power saving [14], which take into account the traffic load, which is based on real-time data traffic and advocates the use of energy-efficient components to improve the energy consumption of the BTS at the hardware level; and (ii) the use of renewable sources of energy as main power sources for BTSs to reduce the electricity bill and the negative environmental impact of conventional fossil fuel-based energy resources due to their carbon footprint [15].

As for the use of renewable energy, sustainable BTSs have become a real solution to the problem. BTS configurations range from standalone solar powered BTSs with storage batteries, to grid-connected ones, in which the grid provides power to the BTS when the photovoltaic system (PVS) and the battery energy storage system (BESS) do not provide enough power. Hybrid configurations can also be found, which integrate different renewable sources such as wind and solar energy along with conventional energy sources such as diesel generators [16,17,18]. The authors in [19,20,21] carried out a detailed analysis of the technical and economic feasibility for different configurations used to power BTSs. However, to the best of the author’s knowledge, there are no research studies which focus on the economic benefit derived from considering a BTS as an MG in terms of optimal energy flow control. Needless to say, achieving energy efficiency needs to go hand in hand with providing uninterruptible and reliable power supply to the critical loads for operation. In this regard, BTSs powered by only solar energy pose a challenge of dimensioning the PV system and the Battery Energy Storage System (BESS) [22]. On the other hand, in grid-connected BTSs, when a power outage occurs, the communication networks should remain operative. Moreover, a BTS based on renewable energy without an operational strategy may increase the OpEx, thereby delaying the return on the initial investment and making this alternative less attractive for network operators. Design of an operational strategy must consider the BTS energy profile in terms of the energy consumed by each component, and thus, the energy allocation per function within the BTS. This topic has been properly researched in [23,24], showing that the radio equipment and amplifiers account for more than 60% of the power consumed; 11% is due to the DC power system, while the cooling equipment is responsible for 25% of the power consumed. Therefore, an optimal design of the radio equipment and the cooling system can reduce the power drawn by the BTS. Likewise, it is very important to develop strategies directed towards BTS Energy Savings, which can be applied to both the radio equipment, e.g., radio sleep mode [25], and to the cooling, e.g., passive cooling, advanced climate control [26] and power electronics. Additionally, the forecasting of the available energy from renewables should be considered due to their stochastic nature, and the electricity price, battery health and lifespan, charging and discharging cycles for the BESS etc. are also required. Hence, an intelligent energy management of the BTS components has to be adopted.

To develop the above-mentioned operational strategy, a microgrid (MG) paradigm can be used to model a BTS, consisting of a low-voltage network (DC and/or AC) that integrates renewable energy sources (RES), a BESS and controllable loads connected to the main grid. To provide the MG-based BTS with intelligent energy management, a local controller and an Energy Management System (EMS) should be implemented. By supervising and coordinating the operation of the sources of renewable energy, the BESS, the controllable loads (critical and non-critical) and the network devices, the BTS performance can be improved.

The operation of an MG-based BTS should be based on a sensing system for determining the value of electrical and environmental parameters. In [27], a comprehensive review of several PV sensing systems is presented. Overall, these systems have either a high cost [28] or reveal limitations regarding the number and type of the parameters to be measured [29]. The practical application of monitoring systems poses many challenges, such as the harsh environmental conditions the different components must withstand and the degradation of the electrical components e.g., connector corrosion, resource restrictions with respect to the energy efficiency, real-time constraints and low-cost implementations, to name but a few [27]. To successfully address these challenges, the monitoring system proposed in this paper is based on up-to-date technologies which guarantee real-time operating conditions, low energy consumption, protection against harsh environmental conditions, firmware OTA updating, local and cloud-based data storage, high computational capacity, accurate measurements and support for several communication protocols, e.g., I2C, SPI, Modbus and MQTT, implementing a low-cost approach. Another strength of this approach is that the system not only monitors the BTS variables but also is able to control the energy flow among the different elements comprising the BTS. This is carried out by an EMS algorithm developed for this application.

The scalable MG-based architecture for BTSs described in this paper includes the following features:Transformation of BTSs in scalable and controllable DC Microgrids to reduce the OpEx. For this purpose, an Energy Management System (EMS) is developed. EMS manages the energy flow in the MG-based BTS based on different scenarios as described in Section 4.The BTS is modeled on the IEC 61850 standard, which improves interoperability and scalability, supporting, in the future, the integration of new BTSs. The equipment used in BTSs are usually manufactured by different vendors overlooking the interoperability between these devices. Thus, a significant advantage of using the IEC 61850 is that it facilitates future extensions.The data collection scheme is based on the cloud from where the information becomes available. This provides always-on, real-time data collecting and the possibility of cloud computing for real-time management.The hardware/software (HW/SW) architecture within the BTS is implemented by using low-cost off-the-shelf hardware. This reduces the time for the return of investment and becomes an economically feasible solution for network operators.

The main contributions of the paper are summarized as follows: (i) the design of a scalable architecture to turning BTSs into Scalable and Controllable DC Microgrids which can be applied to any type of BTS; (ii) the architecture is based on the IEC 61850 standard, which enables the distributed communication among the BTSs; (iii) the low-cost hardware implementation of the system, which decreases the time for return on the investment, making it more attractive for network operators; (iv) the proposed architecture lays the foundations to allow several BTSs to work cooperatively, sharing energy among them and injecting the surplus energy to the grid. In short, it allows a set of controllable MG-based BTSs to be aggregated through a centralized management in the shape of a Virtual Power Plant (VPP) [30,31]; (v) implementation of an EMS which focuses on optimizing the monetary benefits obtained by managing the charging and discharging of a BESS along with the production of photovoltaic (PV) energy and the shedding of non-critical loads; (vi) the development of a smart sensing strategy, as all the sensors have communication capabilities, smart processing based on low-cost hardware and cloud computing facilities for PV forecasting. This allows several BTSs to work cooperatively.

In the literature, there are virtually no works dealing with the transformation of a conventional BTS into a scalable and controllable DC MG. An experimental setup has been developed to validate the approach.

The remainder of this paper is organized as follows. Section 2 describes the conventional BTS layout and the proposed BTS architecture. Section 3 described in detail the proposed MG-based BTS. Results are presented in Section 4. Finally, conclusions are drawn and future work is outlined in Section 5.

## 2. Architecture of the Microgrid-Based Base Transceiver Station (MG-Based BTS)

In this section, the proposed architecture for the MG-based BTS is described. To evaluate the degree of the transformation, the conventional BTS is first introduced.

### 2.1. Conventional Base Transceiver Station

Figure 1 shows the architecture of a conventional grid-connected BTS without renewable energy generation nor local controller.

This BTS layout can be applied to any BTS regardless of the type [9]. As seen in Figure 1, the BTS consists of a grid-connected power supply system which integrates a rectifier, the BESS unit, the Base Band unit (BB), the Radio Frequency (RF) unit, the Power Amplifier (PA) and different AC loads, such as the cooling system and lighting. The DC output of the rectifier is connected to: (i) the BESS, which acts as a backup source to facilitate continuous operation in case of a power outage; (ii) the transmission/reception equipment (RF, PA and BB). The grid-connected power supply system manages the charge of the BESS.

A power modeling approach for conventional BTSs can be found in [32], in which, at the time of publication, it was estimated that, in a full-load scenario, the power demand by BTSs would be reduced by 50% and 20%, respectively, from 2014 to 2020. With this power estimation in mind, the BESS size is determined as a function of the required autonomy, i.e., the amount of time the BESS can power the BTS uninterruptedly to ensure 100% operability in case of a power outage.

### 2.2. Proposed Architecture for the MG-Based BTS

Figure 2 depicts the block diagram of the proposed architecture for a grid connected BTS with solar generation based on a microgrid architecture.

The MG-based BTS consists of the aforementioned BTS along with a PV system with Maximum Power Point Tracke (MPPT) control and a DC bus connected to the BTS rectifier. This DC bus acts as the microgrid main bus to which the different elements are connected. The BESS and non-critical loads are directly connected to the DC bus without DC/DC converters. This is the approach adopted by most companies with the aim of maximizing short-term profitability, since a more technologically sophisticated design would increase the costs. In the BTS architecture proposed in this paper, all loads (critical and non-critical) and connections of the different parts of the MG-based BTS become controllable elements through a set of switches (Figure 2). Furthermore, several variables, such as currents and voltages in loads and the BESS are monitored. Thus, different operating modes and energy flows within the BTS can be managed by designing a local controller.

This local controller is implemented on an Single-Board omputer (SBC) Raspberry Pi 4 model B and some electronics associated. The reason for choosing this SBC is threefold: firstly, a reduction of power consumption is required; secondly, because of its high processing power; finally, the Raspberry Pi platform has been successfully used in similar works [33,34,35]. The associated electronics consists of a main board and a series of latching relays electrically connected to the main board. In the main board, several sensors and conditioning electronics have been included to collect electrical parameters of the installation, such as BESS and load currents, BESS voltage and some ambient parameters (temperature, humidity, etc.). These parameters are sent to the SBC through I2C protocol. The local controller, which also receives the DC bus voltage and PV modules parameters from the MPPT solar charger controller through Modbus protocol, processes all this information to calculate other variables, such as the State of Charge (SoC) of the BESS, which allows the MG-based BTS to be managed. To implement the control of the MG-based BTS, there are also three latching relays which connect or disconnect the BESS and the non-critical loads to the DC bus, and the rectifier to the grid. Henceforth, these associated electronics will be referred as the SBC driver, since it acts as the bridge between the local controller in the SBC and the hardware of the installation.

Likewise, to provide the basis for the smart sensing strategy, a server based on IEC61850 standard has been implemented. This allows the MG-based BTS to be remotely controlled as a VPP node using an external IEC61850 client. Previously, an IEC61850 plant model of the MG-based BTS is proposed to store data in a standard way.

The MG-based BTS has been designed for the worst-case scenario for a macro BTS with a rated power of 3 kW. The experimental setup has been developed using the same RF equipment and power amplifiers as those installed in 3-kW BTSs. The rated power is equally split among the three controllable loads: two critical loads corresponding to the always-on transceivers and a non-critical load for transceivers that can be switched off, and auxiliary equipment, such as the cooling system, when necessary, and lighting. This power configuration can be easily scaled to meet the requirements of more power-demanding BTSs.

## 3. MG-Based BTS Operation

### 3.1. MG-Based BTS Measurements and Control Electronics

In this subsection, the parameters to monitor and the electronics required to accomplish the stated objectives are described. The relationship between the elements that measure the electrical parameters and control the flow of energy in the MG-based BTS is shown in Figure 3. In the following paragraphs, these elements will be explained.

Firstly, the DC currents are measured using the ACS758LCB-100B-PFF-T and the ACS758LCB-100U-PFF-T sensors [36] (Figure 4). The range and the sensitivity of these sensors are ±100 A and 40 mV/A, respectively. Regarding the currents, these values are enough for an expected maximum BESS current of ±60 A and a maximum single load current of 20 A. As for the sensitivity, at a full scale, the sensors will output ideally a DC voltage signal between 0 V and 5 V, covering the dynamic range of the ADC used. The conditioning circuits for the current sensors consist of low pass filters with a cutting frequency of 50 Hz. They are used to eliminate possible coupled noise from the grid. Finally, the typical output noise of the sensor, in measured current units, will be 0.33 A, which is acceptable for this application.

The BESS voltage is measured using a voltage divider to scale the voltage to the dynamic range of the ADC (Figure 4). The ADC used to convert the DC currents and voltage measurements is the ADS1115 [37]. This low-cost four-channel 16-bit I2C ADC has an input range from 0 V to 5 V, a maximum sampling frequency of 860 Hz and features low power consumption. Furthermore, to measure the ambient parameters (temperature, humidity and pressure), the I2C BME680 sensor [38] is used due to its low cost and low power consumption.

Once the measurements are obtained, the BESS SoC is estimated. This is an important parameter for the management of the MG-based BTS. In [39], a complete state of the art analysis of different algorithms for lead acid batteries SoC estimation can be found. Among them, the current integration or Coulomb counter method has been used in this paper. This method, along with the manufacturer’s specifications of the batteries, are merged to provide an accurate estimation of the SoC, through the floating voltage of the BESS and the dynamic capacity depending on the discharge current rate [40].

The MPPT solar charger provides several parameters related to the PV installation through the Modbus protocol [41]. For the sensing strategy proposed in this paper, the following parameters are required: (i) the output voltage of the MPPT solar charger; (ii) the output current provided by the charger; and (iii) the voltage, current and power of the PV modules. All these parameters are collected in the SBC.

The loads are connected or disconnected through low cost off-the-shelf MOSFET- CSD18536KCS 60 V N-Channel transistors [42]. Their maximum drain-source voltage is 60 V, over the maximum voltage of the DC bus 58 V, which is imposed by the rectifier of the RF equipment at the dedicated output for the BESS [43]. Furthermore, for the connection of the different parts of the installation, EW60-1A3 12VDC 60 A [44] latching relays have been included in the hardware architecture. Figure 5 briefly depicts the conditioning circuits of the transistors and relays of the SBC driver.

Regarding the power supply for the electronics, there are three isolated sources: (i) the SBC power supply; (ii) the power supply for the main board of the SBC Driver and the transistor-based switches for the critical loads; and (iii) the power source for transistor-based switches for the non-critical loads. This configuration breaks ground loops (leading to less interference in data wires), isolates low power electronics from the higher power parts (cutting off unexpected overloads) and also isolates the SBC and its peripherals, which avoids problems related to different grounding configurations among devices. Regardless of the source, the power is always drawn from the BESS to avoid the eventual loss of power due to a grid failure. To this aim, three different low-cost DC/DC converters, with a rated current of 4 A that meets the current demanded by the SBC driver and the SBC, have been used.

To keep the isolation among the aforementioned parts, drivers based on the FOD3182 optocoupler [45] and the digital isolator MAX14937 [46] for the transistors and I2C communications are included. The SBC controls the optocoupler drivers through two I2C expanders [47].

Finally, electromagnetic noise, due to abrupt changes in DC current flows, must be kept within certain limits. In the early stages of the MG-based BTS development, noise coupling was detected in measurements, generated by the DC currents of the BESS and the loads. Consequently, an analysis of this electromagnetic interference was performed, seeing that a 100-mV noise was coupled into the input signals of the ADC (leading up to a 5 A error in the measurements of DC currents through the loads) and into the BME 680 power supply (leading to a malfunctioning). In the final design of the SBC driver, the components were rearranged to minimize the coupling noise. Figure 6 depicts current measurements taken in the final set up of the SBC driver in the presence of electromagnetic interference. Figure 6 shows a sudden change in the currents through the non-critical loads (Figure 6a) and in the BESS (Figure 6c). Particularly important is the 40 A change in the current through the BESS, which induces electromagnetic interference in the input signal of the ADC, introducing a 0.4 A measurement error in the critical load DC currents. This error is shown in Figure 6b in the shape of a noise peak. Hence, it is important to reduce the negative impact of the noise on the ADC input, achieved in the final design.

### 3.2. Wireless Weather Station

A wireless weather station has been developed and installed close to the PV modules. The architecture of the wireless weather station is shown in Figure 7. The main purpose of this weather station is to collect environmental data, which will be used in the smart sensing strategy focusing on the prediction of PV generation as part of the VPP control strategy.

The weather station is made up of two modules: (i) the sensing module, which consists of a set of sensors used to measure environmental parameters related to the PV modules, such as irradiance, wind direction and speed, and PV module temperature; and (ii) the acquisition electronics based on an ESP32 microcontroller and a BME 680 sensor. Regarding the sensing module, a SR05 pyranometer [48] is used to measure the horizontal irradiance [49]. This sensor provides an analog voltage output ranging from 0 V to 2 V for measured irradiance values from 0 to 2000 W/m^2^, respectively. The wind direction and speed are obtained by a Davis Instruments anemometer with an operating range from 1 to 322 km/h for wind speed with a resolution of 1 km/h using a sensor based on a reed switch whose output is directly connected to the microcontroller. For wind direction, the operating range goes from 0° to 360° with a resolution of 1 [50]. Finally, the PV module temperature is measured by a PT100.

Figure 8 shows the PT100 fixed to the PV module, the anemometer with PV modules at the background and the acquisition electronics of the weather station.

As far as the second module is concerned, the ADC ADS1115 converts the voltage information from the pyranometer and the anemometer (wind direction) to digital values which are sent to the microcontroller through the I2C protocol. Additionally, the BME680 sensor measures and processes the ambient temperature, humidity and pressure. This information is also sent to the ESP32 microcontroller via an I2C protocol. A fan has been installed to keep the ambient temperature of the electronics stable and minimize any fluctuation in the temperature measured by the BME680.

The two-core ESP32 microcontroller software architecture is based on the real-time operating system FreeRTOS, which allows to allocate tasks in a particular microcontroller core. One core executes a task which reads the sensor measurements. The other core is responsible for sending the information to the SBC via MQTT (more details will be given in the next subsection), and for sending the information to the IoT cloud of Matlab, named ThingSpeak.

The variables stored in Thingspeak (panel and environment temperature, humidity, irradiance, wind speed and wind direction) are shown in Figure 9.

### 3.3. Communication System

The communication system is the basis of the smart sensing operation. It is divided into the local communication architecture, which is aimed at creating communication channels between the different SBC processes, the SBC driver and the weather station, and the global communication strategy, which allows the SBC and a global controller to communicate via the IEC61850 standard. A specific extension of this standard is the IEC61850-7-420, which defines the communication and control interfaces of Distributed Energy Resources (DERs) and proposes logic nodes (LN) to completely describe DERs and control systems associated to them. This extension can be used to model communications in MGs [51].

The local communication architecture is based on “The Robot Operating System” (ROS). ROS is an open-source operational system mainly meant to develop robotic systems [52]. In the local communication architecture, the SBC driver sends the measurements via I2C and Modbus to the processes in the SBC. The SBC collects and publishes the measurements as ROS topics to be shared by all the processes. On the other hand, the weather station sends the environmental information via Wi-Fi through MQTT protocol to the SBC. The choice of the MQTT protocol is due to its stability. Finally, the environmental parameters are transformed into ROS topics at the SBC. The block diagram in Figure 10 shows the communication protocols of the smart sensing system.

As for the global communication strategy, the EMS and the IEC61850 Server read the ROS topics related to the measurements and update the IEC61850 plant model parameters of the MG-based BTS. As a result, the IEC61850 clients can directly access the MG-based BTS parameters through a standard communication channel via a TCP/IP protocol. Furthermore, this global communication strategy can receive control instructions from global controllers to change the behavior of the local controller or to directly control the MG-based BTS. The IEC61850 communication standard provides high scalability and interoperability allowing the MG-based BTS to be easily extended with any system or equipment which comply with the standard. From a VPP perspective (based on BTSs), this standard brings an economic benefit, since after the initial investment in the development of an MG-based BTS, the time and capital cost of adding a new MG-based BTS is dramatically reduced.

Figure 11 depicts the plant model of the MG-based BTS, which consists of different logical devices (LD), each one representing one component or device with its own entity. For instance, the logical device labeled as LD RGL represents the MPPT solar charger. This LD contains the ZRGL class, which has been fully specified in this work and cannot be found in the standard. It can be considered that the ZRGL class (Appendix A) expands the IEC61850 (specifically IEC61850-7-420), as the standard does not include any class to describe a DC/DC voltage regulator.

### 3.4. Processing and Energy Management Systems

#### 3.4.1. Processing System

As introduced in the general overview, the local controller is based on an SBC, which implements the local controller and an HMI (Human-Machine Interface). The local controller consists of a set of processes responsible for coordinating and controlling the different operating modes of the MG-based BTS. Among its main functions are collecting measurements from the SBC driver, the SoC estimation, the control of the energy flows through the EMS, and adding high-level features of the IEC61850 standard. A global controller in the shape of an IEC61850 client could perform control actions and monitoring tasks. Finally, the HMI provides an integrated interface, which displays the data collected from the SBC driver and control variables with the aim of facilitating the manual intervention in the MG-based BTS operation.

The processes comprising the local controller and the HMI can be categorized in three levels: (i) the physical interaction level in which the data from the sensors and the BTS parameters is obtained; (ii) the logic level which is based on the developed IEC61850 Server and implements the EMS, the high-level functionalities, and the modification of the IEC61850 plant model; and (iii) the HMI.

The processes are executed at fixed time intervals depending on the level they are in and the data dependency among them. Those processes in the physical interaction level and those implementing the HMI have the smallest execution interval (2 s) for efficient operation, since the HMI must display the data collected by the processes in the physical interaction level. The execution interval for the processes in the logic level is set to 5 s. It is important to note that the Raspberry PI OS does not feature real-time capabilities. Therefore, the definition of the execution intervals depends on the process execution time.

#### 3.4.2. Energy Management System

The proposed EMS is focused on optimizing the monetary benefits obtained by managing the charging and discharging of the BESS in conjunction with the generation of PV energy and the management of non-critical loads. To implement this strategy, it is necessary to participate in an electricity market with a two-tariff pricing scheme. These markets are common in many countries, as this pricing scheme encourages the consumption of energy in periods where the energy demand is lower. This is the case on the Spanish electricity market [53]. In this type of billing, prices are significantly more expensive for the peak time (PT) tariff in comparison the off-peak (OT) tariff. This type of pricing scheme is particularly recommended for the case under study for two reasons: (i) the PV energy generation takes place mostly during the PT period, which encourages self-consumption at times when the price of energy is higher; and (ii) the energy consumption in the BTS does not tend to vary greatly during the day and the average daily price of energy with two periods is usually lower than the default tariff, which makes the total price of energy consumed by the BTS lower in the case of a two-tariff pricing scheme [54].

In the proposed EMS, it is considered that the installed PV power is less than or equal to the one consumed by the loads installed in the BTS, since the installed equipment does not allow grid feeding. Nevertheless, this strategy can be extended by considering a surplus of PV that can either be injected into the grid or used for battery charging. In this work, it is also assumed that the BESS is working at the proper temperature thanks to the cooling equipment.

Regarding the BESS size, in this work, only the back-up batteries previously installed in the BTS are considered. A further increase in the BESS size could be feasible for new BTS projects, in which more efficient storage technologies, such as Lithium-ion, can be implemented, or if the current battery prices decrease. Since the back-up batteries are used in the EMS strategy, they will not always be fully charged in case of a power outage. To overcome this drawback, three actions are taken:It is always guaranteed that the BESS discharge, scheduled by the EMS, does not exceed a minimum level so that the power support is available in the case of an outage.The BESS is charged, after a discharge process, at the beginning of the off-peak tariff period when the electricity price is low. This increases the number of hours during the day in which the BESS is fully charged.In the event of a power outage, non-critical loads are disconnected to maximize the back-up time provided by the BESS.

It is possible to obtain multiple operating states by using the installed switches (see Table 1). The following states provide an optimal solution for the operation of the BTS while at the same time making an efficient use of the BESS:State 0 or Back-up State: this state comes into operation when there is a drop off in the main grid. In this state, which rarely occurs in countries with reliable grids, the consumption of the installation is reduced to only the critical loads, and the BTS is powered by the BESS and the energy available from the PV modules at that time.State 1 or Transition State (Peak Tariff) or Battery Charging State (Off-Peak Tariff): this state is used as a transition state in the case of working in the Peak tariff period. The BTS remains in this state for a maximum of 30 s. During the Off-peak tariff period, this state is used to charge the battery.State 2 or No Battery State: in this state, the BESS is disconnected, either because it has already been charged to the desired level in the Off-Peak tariff period or because it has been discharged to the defined level in the Peak-tariff period.State 3 or Battery Discharging State: the BESS is discharged by powering either part or all of the non-critical loads. Thus, an appropriate discharge current can be selected considering the characteristics indicated by the manufacturer. The remaining loads in the BTS are fed from the grid and the PV system.State 4 or Island State: in this state, the BTS works in island mode without drawing power from the grid. This state is used when the production of PV system is sufficient to power the whole BTS, supported by the discharge of the battery within the appropriate discharge range.State 5 or Cloud State: this state is used to avoid unnecessary changes of state produced by the drop of PV power occasionally caused by a cloud, while protecting the BESS by keeping it within proper discharge ranges. If the PV power falls abruptly and the BTS is working in State 4 or Island State, the non-critical loads are disconnected, and the average PV production of the last few minutes is continuously checked. If this average PV power generation continues to decrease in the following minutes and the PV production does not recover, the system returns to a grid-supported state.

Once the operational states have been introduced, the proposed finite state machine (FSM) representing the behavior of this EMS is described. Three levels of priority are established, in the state transition:(1)Very High Priority: In the event of a grid outage, the state is immediately changed from any state to State 0 or Back-up State. When the outage is over, the FSM enters State 1 or Transition State.(2)High Priority: If there is a change from the off-peak tariff period to the peak tariff period or vice versa, there is a transition from any state to State 1 or Transition State.(3)Normal Priority: Common EMS operation with grid available and operating within one of the working periods. The transitions in this mode are described in the following table.

To gain an insight into the proposed EMS, a state diagram flowchart describing the operation of the FSM in Normal Priority is shown in Figure 12. To reduce the clutter, the Very High Priority and High Priority transitions are not depicted because they follow basic rules.

The green color shows the states that are used in both PT period and OT period, whereas the orange color shows the states that are used only in PT periods. Finally, the yellow color shows the back-up state that is used whenever there is a grid outage.

Once the different transitions have been described, it is possible to analyze the BTS operation modes with the implemented strategy.

When a Very High Priority event occurs, i.e., the grid outage, the system automatically enters the Back-up State. In this state, the non-critical loads are disconnected, and the entire system is powered by the BESS and, if available, by the PV power. This state is maintained until the grid is operative again.

The next condition to be checked is the High Priority event, which occurs twice a day: once when shifting from the PT period to the OT period and the other when changing from the OT period to the PT period. In this case, the EMS operating mode changes completely, as described below in the normal priority operate mode.

The EMS operates in the normal priority mode most of the time, as the very high priority mode only takes place when a grid outage occurs and the high priority mode two moments a day. During the OT period, the BESS is charged to the desired SoC level, which is set by the designer according to the BESS characteristics. Once the BESS is charged, it goes into a standby mode. Conversely, during the PT period, the BESS is discharged in an appropriate manner, considering their characteristics, to guarantee suitable discharge currents and levels to prolong its lifespan and optimize its total capacity [55]. At the same time, PV production is considered in order to choose the moments when it is appropriate to use the island mode of operation in which PV and BESS are responsible for powering the entire BTS by disconnecting it from the grid. It is important to note that all state transitions in this mode of operation are made by applying hysteresis to the PV thresholds of state switching to ensure as few transitions as possible, thus avoiding high-frequency state changes. Following this approach, the Cloud State is implemented to prevent that if a drop in PV production is produced by an occasional cloud, no reconnection to the grid takes place.

This strategy is aimed at optimizing the management of the BESS by ensuring that the discharge currents of the BESS are within the parameters set by the manufacturer and that the depth of discharge chosen maximizes the relationship between the number of BESS cycles and the capacity of the BESS [55].

As an extension of this EMS, in installations where more PV power is installed than the amount of load demanded by the BTS, a new battery state could be considered where the BESS is also charged during the PT period with the surplus of PV. To this end, the BTS PV production forecasts [56] could be used to calculate the periods during the day when this surplus could be produced and, in this way, support the OT charging strategy.

To realize the economic study of the savings obtained with the implemented EMS, it is essential to know the price differential between the PT and OT, as well as the BESS characteristics: battery efficiency, optimal depth of discharge and number of life cycles. For a PV system, it is fundamental to determine the amount of power generated according to the location of the installation as well as the prices in the production hours. In Section 4.4, a study is carried out for the specific case study, also obtaining general conclusions for any market and location.

### 3.5. Interface System—HMI

The aim of the HMI (Human-Machine Interface) is to display the collected data from the MG-based BTS and the control variables. It also allows the manual operation over the MG-based BTS to be performed. This interface continuously communicates with the local controller to coordinate the operation of BTS and the data monitoring. The HMI has two operational modes: (i) operator mode aimed at manually manipulating all controllable variables of the MG-based BTS and taking measurements without using the local controller processes; and (ii) normal mode, which just acts as a graphical interface to display all the information compiled by the controller and, therefore, by the MG-based BTS. The HMI is implemented using Node-RED, an open-source software which provides a web-based dashboard facilitating its use for any device inside the same network. Figure 13 shows the HMI’s appearance.

## 4. Results

### 4.1. Experimental Setup

For the experimental setup, an MG-based BTS has been developed with a rated load power of 3 kW, using the same RF equipment and power amplifiers as those installed in 3-kW BTSs. The sizing of the PV system must consider the space available at the BTS site. Nevertheless, a solar charger controller with MPPT and rated power of 2.7 kW for a BESS of 48 V/190 Ah (composed of four lead batteries) and nine 300-Wp PV panels [57] in a 3 × 3 configuration (3 kW PV power peak) are used in the experimental setup. The charger controller, the BESS and the non-critical loads are connected to the DC bus as seen in the Section 2.2 and shown in Figure 2.

The tests have been performed with a maximum of two critical loads (2 kW) and one non-critical load (1 kW). Figure 14a shows the interior of the main cabinet of the MG-based BTS and Figure 14b shows the PV installation including the meteorological station.

### 4.2. Use Case

In this section, the results obtained with the plant described in the previous section are shown. The first step is to set the design parameters of the EMS according to the characteristics of the plant and the prices of the Spanish energy market in winter time: PT, period from 12:00 to 22:00; OT, period from 22:00 to 12:00; $SOC_{min}$, 35%; $SOC_{max}$: 95%; $TH_{PV}$, 1900 W; $TH_{PV}^{-}$, 1800 W and $TH_{PV}^{+}$: 1950 W. The OT and PT values are set by the Spanish energy market. The $SOC_{max}$ and $SOC_{min}$ are based on the battery datasheet provided by the manufacturer, which sets an optimum DoD of 60%. The PV threshold is also defined considering the battery datasheet, stating that the power provided by the BESS is always below 2 kW and most of the time around 1 KW or less. A hysteresis value of 100 W for the PV power is also set to avoid frequent state transitions which could be caused by small oscillations of PV.

Once the design parameters have been described, Figure 15 shows the results obtained in a 24-h test. The test starts at 6.00 a.m., on the 22nd of December 2020, to facilitate the comprehension of the experiment, since it begins in a state in which the BESS is already fully charged to the level defined by the $SOC_{max}$ parameter.

Figure 15a presents the hourly energy prices. It can be seen that the PT period lasts from 12:00 to 22:00 h and the OT period from 22:00 to 12:00 h. The strategy designed takes advantage of the electricity price difference between the two periods, which is around 0.6 €/kW.

Figure 15b shows the generation of PV power, on a mostly sunny day with different types of clouds to demonstrate the potential of the algorithm. The PV power is shown in blue, the 15-min average PV power in yellow and the irradiance is depicted in red. Analyzing the correlation between the irradiance and PV power measurements, it can be seen that the system is tracking maximum power at all times. The effects produced by the clouds and the potential to use the average PV over the last 15 min is described below, including a zoom of the figure in this working area, Figure 16.

Figure 15c shows the power consumed by the loads, the power drawn from the grid and the power flow from/to the BESS. Negative values for the power represent consumption and positive values represent supply. The power consumed by the critical loads are shown in blue. These loads have a rated power of 2 kW and are always connected to the power supply. The power consumed by 1-kW non-critical loads are shown in red. These loads are disconnected when the EMS enters the State 5 or Cloud State, which is used to keep the installation in island mode while protecting the maximum discharge current of the BESS. Finally, the power provided to or withdrawn from the BESS is depicted in yellow, and the power drawn from the grid in purple. As expected, the installation consumes all the energy provided by the PV system, with the support of the grid and the BESS when required.

The BESS charging and discharging strategy is represented in Figure 15d, which shows the BESS SoC. It can be appreciated that the BESS is discharged to the desired level, $SOC_{min}$, 35%, in the first hours of the PT period, from 12:00 to 18:10. During this time, the BESS powers part of the installation, supporting the PV and the grid supply. This minimizes the power drawn from grid in the period of time when the energy is more expensive, reaching the stage where the installation works without drawing power from the grid, which happens when the PV exceeds the 2 kW zone. Then, the BESS is charged from the grid to the desired level $SOC_{max}$, 95%, in the OT period when the energy is cheaper, from 22:00 to 04:45, to be ready for the next day.

Finally, in Figure 15e, the graph representing the states of the EMS is detailed. In the initial part of the experiment the system is operating in State 2 since the BESS is fully charged. The EMS remains in this state until 12:00 when the tariff changes from OT to PT. Furthermore, the graph shows how the BESS is discharged from 12:00 to 18:10, thereby reducing the power drawn from the grid. In this case, the EMS goes through several states, which are described in detail in Figure 15. From that moment, the system returns to the No Battery State or State 2, as it was discharged to the predefined level. At 22:00 with the change from PT to OT, the BESS is charged taking advantage of the lower prices, State 1 from 22:00 to 04:45.

As mentioned above, in order to describe more precisely the central part of the day in which most states are involved, in Figure 16 a zoomed-in section of Figure 15 with the results between 11:30 and 15:30 h is showed..

At 12:00 h, there is a tariff change from OT to PT, and since the PV power generated is greater than 1900 W, the system is working in island mode using the energy stored in the BESS (states 4 and 5). It can also be seen in the graph that between 12:00 and 13:45, the system, taking advantage of the Cloud State or State 5, is capable of maintaining itself in island mode, while protecting the BESS thanks to the disconnection of the non-critical loads when the occasional crossing of a cloud is detected. This is done by working out the 15-min average PV power in the State 5, which avoids occasional PV fluctuations, which cause high frequency changes from island mode to grid mode, while ensuring that the system does not stay for an excessive amount of time in State 5 in which the BESS may have to assume 2 kW of load and non-critical loads are disconnected.

On the other hand, when the PV power is not sufficient to guarantee island mode all the time, the system relies occasionally on the grid so as not to force the BESS to maintain powers greater than 1 kW for long periods of time caused by longer cloud sky. This parameter results in a trade-off and is modified according to the amount of signal that is integrated to compute the PV average, in this case 15 min. This situation is presented from 13:45 to 14.10, when there is support from the grid.

Finally, from 14:10, the PV power generated is not enough for island mode and the system is maintained in State 3 until the BESS reaches the SoC value of 35%. During this state, the BESS powers the 1 kW non-critical load independently, while the PV system and the grid power the critical loads, 2 kW.

### 4.3. Smart Sensing Operation

As has been described, all the sensors include communications capabilities, smart processing based on low-cost hardware and cloud computing facilities for predictions and cooperative work among different BTSs.

Going one step further, a set of controllable MG-based BTSs can be aggregated through a centralized management in the shape of a Virtual Power Plant (VPP), which represents a controllable portfolio of BTSs. Consequently, from a VPP perspective, each MG-based BTS is seen as an aggregated controllable VPP node which can interact with other VPP nodes, i.e., other BTSs, with the aim of facilitating the integration of the BTSs into the grid. Therefore, a hierarchical network structure with a hierarchical control strategy involving both concepts, i.e., microgrid and VPP, can constitute a feasible solution to the challenge of coordinating several BTSs to improve performance.

To allow this cooperative operation mode, the sensors included in the MG-based BTS could provide the EMS system with irradiance predictions in order to better adjust the load and the BESS connection strategy [56]. These predictions are possible because the data provided by the different sensors is stored and processed both locally and in the cloud, thus constituting an intelligent sensor strategy. Additionally, in installations where the PV power installed is greater than the amount of load demanded by the BTS, based on power predictions, a new battery state could be considered where the BESS is also charged during the PT period with the energy surplus from the PV system.

### 4.4. Economic Study

To carry out an economic study, a general methodology is developed to characterize the savings provided by the proposed strategy in different electricity markets, for different batteries and PV power installed. The most important parameter is related to the market and is based on the average price differential between the PT and OT for battery savings and the energy prices during the hours of PV production.

To make the specific calculation, the Copernicus Atmosphere Monitoring Service [58] is used to calculate the power and the moments when the PV energy is produced at the location of the BTS. By performing these calculations for the last year, savings of 153.93 €/kW of installed PV power are obtained; the total savings are 461.78 € for the 3 kW PV installed. Considering an installation price of 1000 €/kW installed, the PV installation return on investment occurs after 6.5 years.

In the case of the BESS, the calculations are based on the datasheet provided by the manufacturer, which sets a battery discharge efficiency of 85%, an optimum Depth of Discharge (DoD) of 60% and 1500 cycle life for the battery used in the BTS. In addition, it is assumed that these batteries are normally changed every eight years (2920 days); since a cycle is carried out every day, a factor of 0.513 (1500/2920) is applied. This way, the number of days per year that the strategy can be applied can be computed, to match the end of BESS life with the time when the BESS would be replaced. Taking into account that the Spanish average daily price differential between the charging and discharging hours of a whole year is approximately 0.09 €/kWh, the annual savings obtained with a BESS with the described characteristics is 16.8 €/kWh of installed capacity. If the correction factor for the batteries to last eight years is considered, the savings obtained is 8.63 €/kWh for each of the eight years. The total annual savings for the installed battery capacity, 9.12 kWh, is 153.26 €/year without the correction and 79.73 €/year after applying the factor.

Finally, Table 2 provides a summary of the obtained results for a BTS with 2 kW of critical loads and 1 kW of non-critical load with energy supplied by the Spanish energy market, considering a daily price differential between the peak and off-peak hours of approximately 0.09 €/kWh.

## 5. Conclusions and Future Work

This paper introduces a new BTS HW/SW architecture based on an MG paradigm, which allows an efficient energy management strategy to be implemented through a local EMS. The key function of the EMS is to derive profit from self-consumption of photovoltaic energy generated on the BTS site. With this aim, the EMS implements a load-shedding approach, which depends on the available energy and is applied to non-critical loads. Furthermore, a controllable BESS is used, in which the charging and discharging cycles are optimized to increase its lifespan. Thus, the power supply of conventional BTSs has been entirely transformed into a more sustainable solution by adding new HW/SW elements, which have been described in detail, namely: (i) a monitoring system for determining the value of several electrical and environmental parameters; (ii) electronics to control the energy flow; (iii) a local EMS system; (iv) an IEC 61850 compliant model for the BTS; and (v) a wireless weather station. The proposed HW/SW architecture has been experimentally validated on a pilot BTS plant subjected to different test and weather conditions. Finally, for a two-tariff pricing scheme which is usually offered by energy providers in the Spanish electricity market, annual savings of 16.8 €/kW per installed battery power can be obtained.

A benefit of this system is that it can be applied to both new and existing installations, providing a two-way connection to the electricity grid, photovoltaic generation, smart measurement systems and the required management software, all integrated in a flexible and low-cost HW/SW architecture. More importantly, by adopting an MG-based model, the transformed BTS can also be regarded as an aggregated controllable VPP node. This greatly facilitates the integration of several BTSs into the grid, thereby improving performance by developing a hierarchical network structure based on a hierarchical control scheme in which both the MG and VPP approaches are adopted. Therefore, within the framework of the MG-based BTS architecture proposed in this work, the feasibility of injecting energy into the AC network can be demonstrated by implementing more complex EMS algorithms within larger microgrids for optimal battery management, relying on meteorological information to forecast PV power generation. These are the new challenges which the authors are currently addressing.

## Figures and Tables

**Figure 1 sensors-21-01202-f001:**
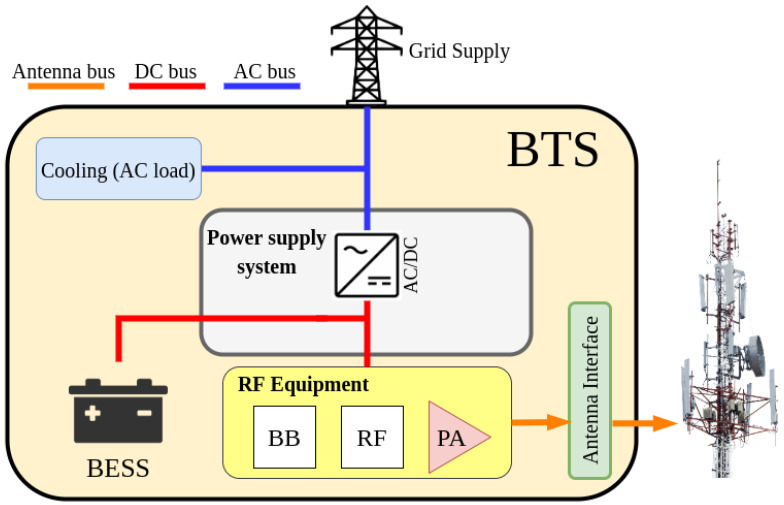
Conventional BTS architecture.

**Figure 2 sensors-21-01202-f002:**
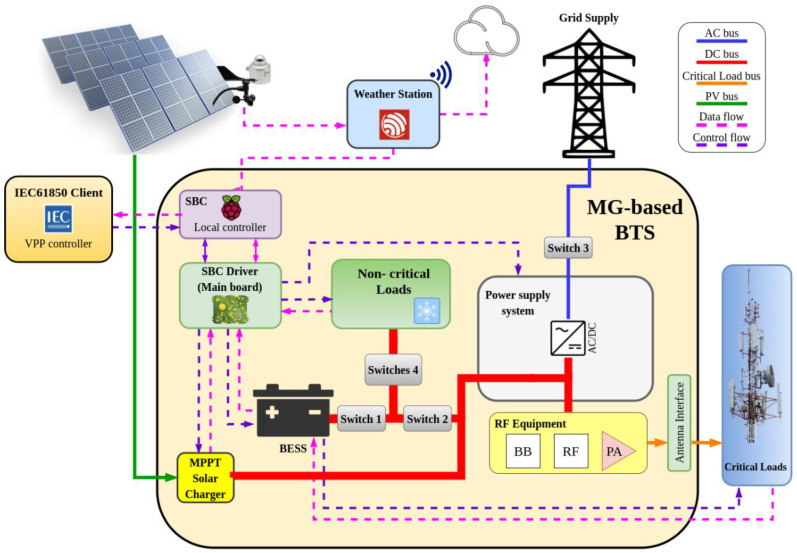
Base Transceiver Stations transformed into DC Microgrids.

**Figure 3 sensors-21-01202-f003:**
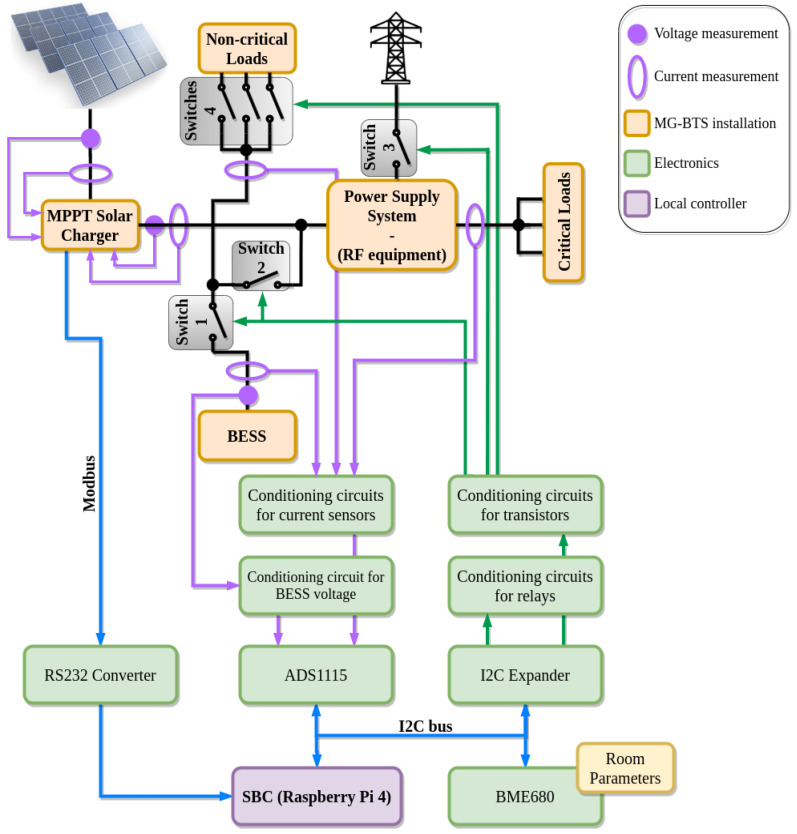
Measurement and control electronics of the MG-based BTS.

**Figure 4 sensors-21-01202-f004:**
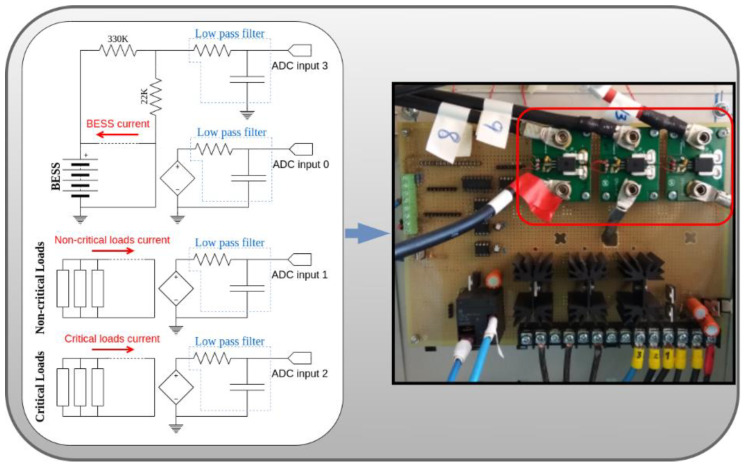
Electronics for the measurement of currents and voltages in BESS, non-critical and critical loads.

**Figure 5 sensors-21-01202-f005:**
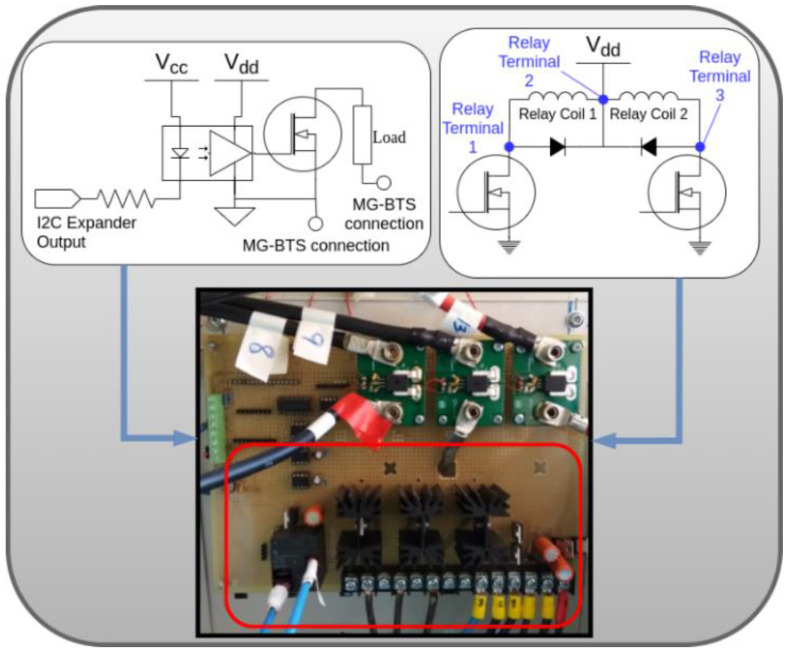
Conditioning circuits of the transistors and relays (SBC driver).

**Figure 6 sensors-21-01202-f006:**
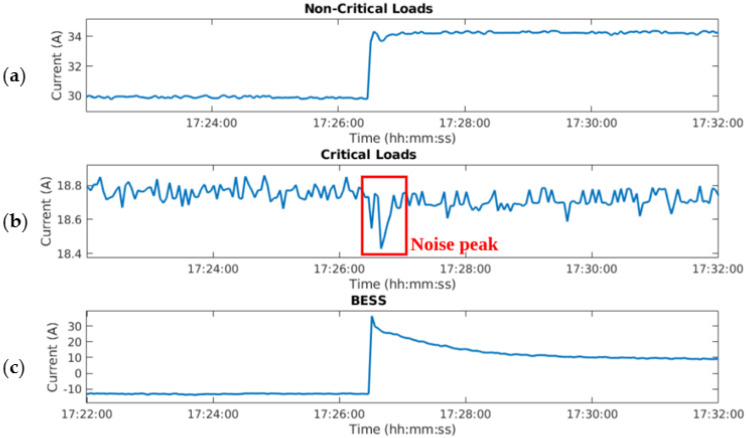
(**a**) sudden change in the currents through the non-critical loads; (**b**) measurement error caused by an approximately 40 A change in the DC current of the BESS; (**c**) sudden change in the currents through BESS.

**Figure 7 sensors-21-01202-f007:**
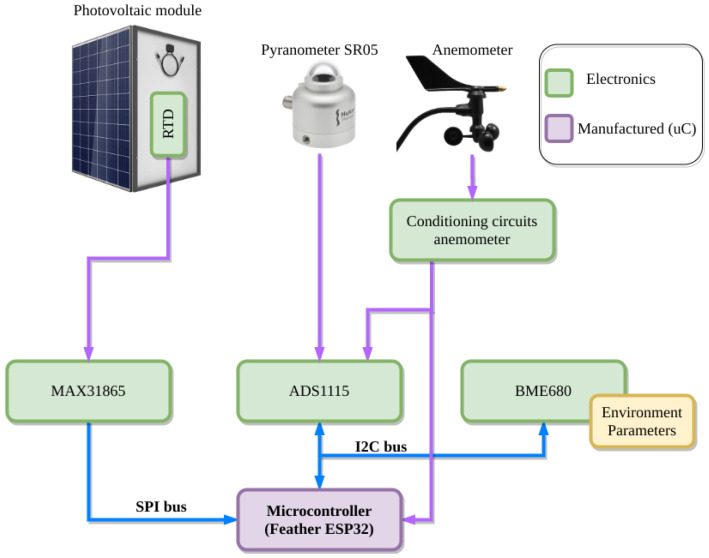
Wireless weather station architecture.

**Figure 8 sensors-21-01202-f008:**
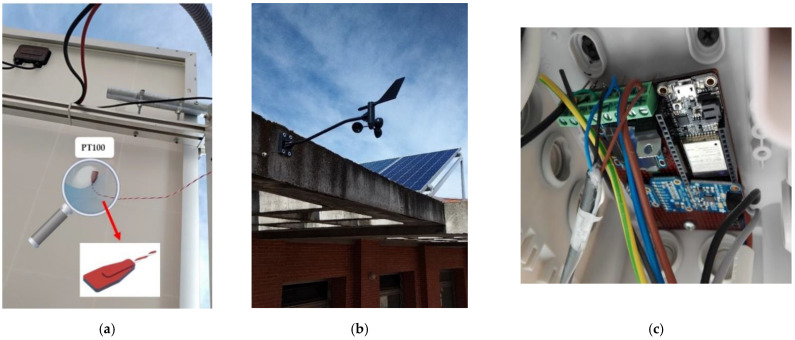
(**a**) Wireless weather station: PT100 fixed to the PV module, (**b**) anemometer and (**c**) the acquisition electronics.

**Figure 9 sensors-21-01202-f009:**
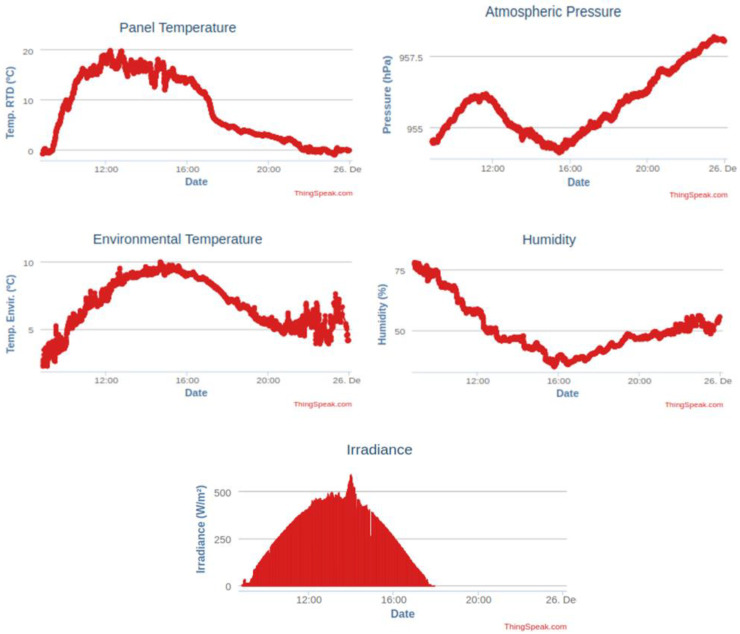
Interface of Thingspeak to visualize stored variables: panel and environment temperature, humidity, irradiance, wind speed and wind direction.

**Figure 10 sensors-21-01202-f010:**
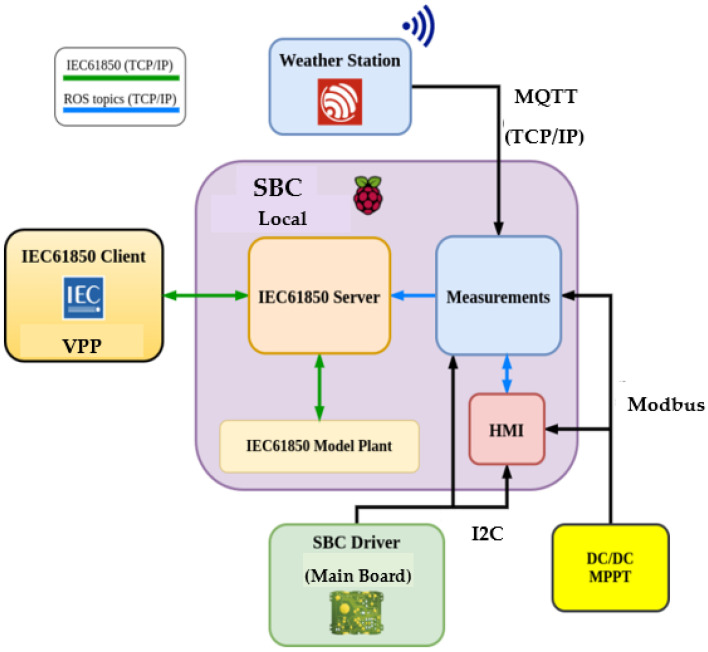
Communication protocols of the smart sensing system.

**Figure 11 sensors-21-01202-f011:**
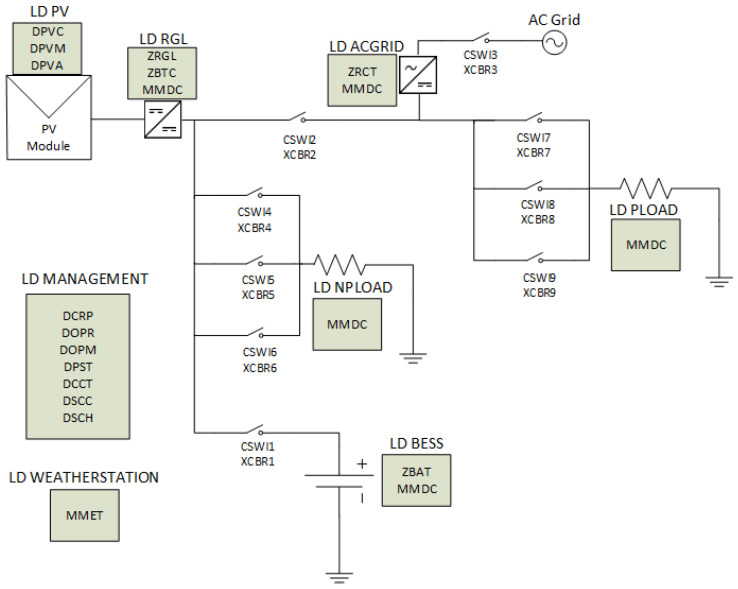
Plant model of the MG-based BTS.

**Figure 12 sensors-21-01202-f012:**
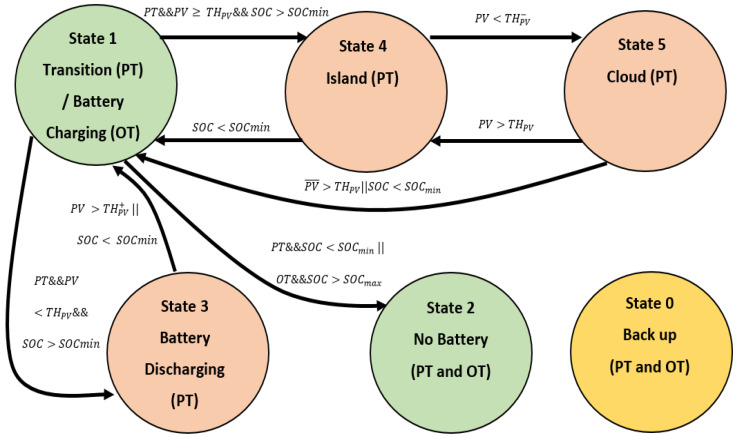
EMS State Diagram showing all states and all Normal Priority transitions.

**Figure 13 sensors-21-01202-f013:**
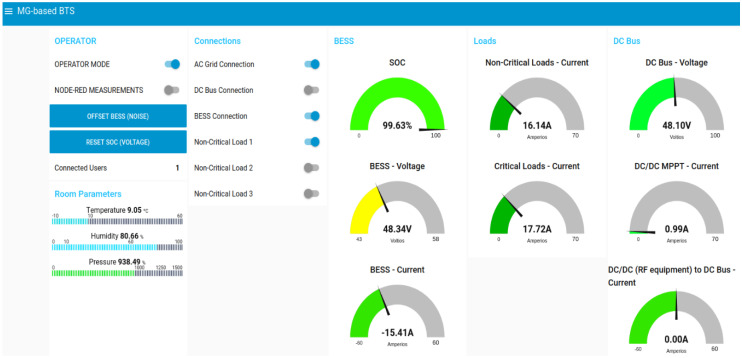
Human-Machine Interface (HMI) designed to monitor and control the MG-based BTS.

**Figure 14 sensors-21-01202-f014:**
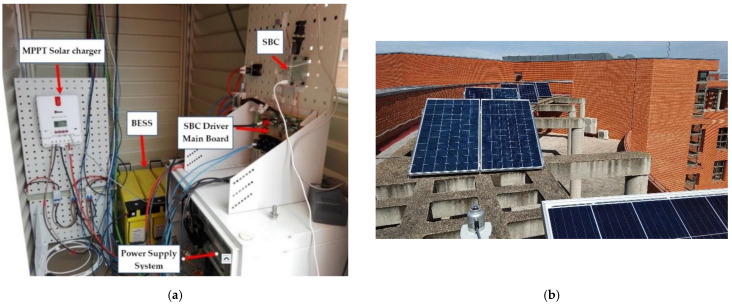
(**a**) Cabinet of the MG-based BTS and (**b**) PV installation including the meteorological station.

**Figure 15 sensors-21-01202-f015:**
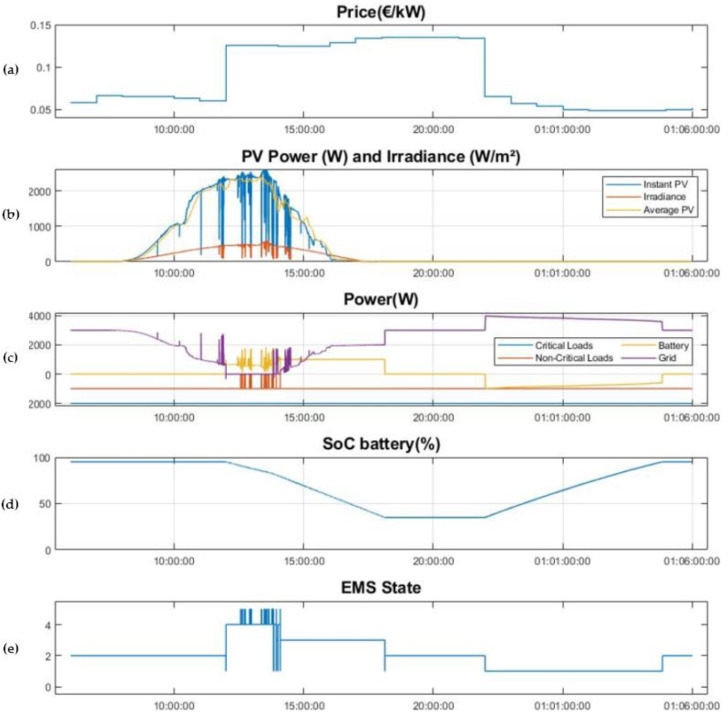
Use Case: (**a**) energy prices in the market for the day under study; (**b**) Instant Power PV production, Average power PV over the last 15 min and irradiance; (**c**) Power consumed by critical and non-critical loads, power consumed and delivered by the BESS and power delivered by the grid; (**d**) SoC level of the battery; (**e**) EMS states.

**Figure 16 sensors-21-01202-f016:**
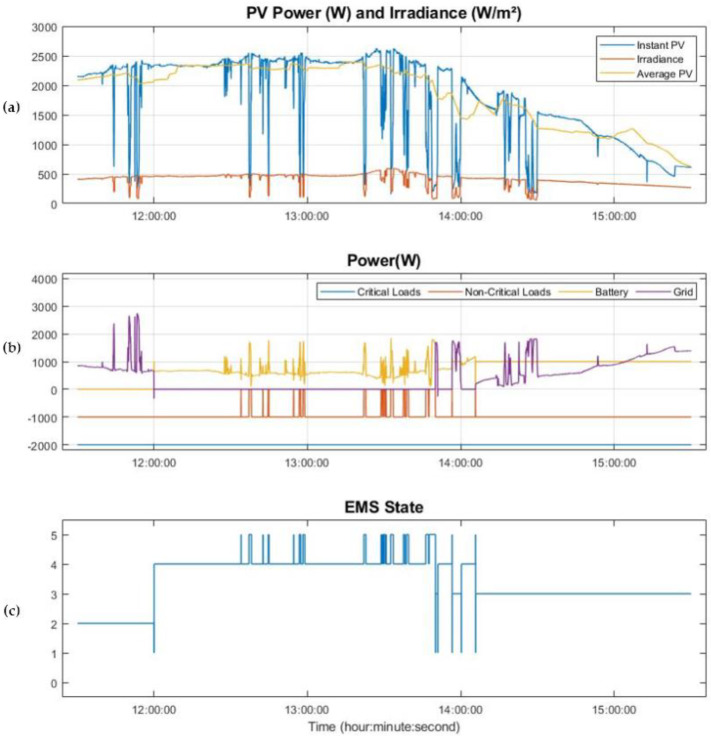
Zoom of use case in the tariff change from OT to PT. (**a**) Instant Power PV production, Average power PV over the last 15 min and irradiance; (**b**) power consumed by critical and non-critical loads, power consumed and delivered by the batteries and power delivered by the grid; (**c**) EMS states.

**Table 1 sensors-21-01202-t001:** Finite state machine transitions in normal priority.

State	Switch			Non-Critical Loads	Transition (Normal Priority)	
	1	2	3		Condition	State
0	ON	ON	ON	OFF		
1	ON	ON	ON	ON	$PT\&\&SOC<SOC_{min}\;||\;OT\&\&SOC>SOC_{max}$	2
					$PT\&\&PV<TH_{PV}\&\&\;SOC>SOC_{min}$	3
					$PT\&\&PV\geq \;TH_{PV}\&\&\;SOC>SOC_{min}$	4
2	OFF	ON	ON	ON		
3	ON	OFF	ON	ON	$PV>TH_{PV}^{+}\;||\;SOC<SOC_{min}$	1
4	ON	ON	OFF	ON	$SOC<SOC_{min}$	1
					$PV<TH_{PV}^{-}$	5
5	ON	ON	OFF	OFF	$PV>TH_{PV}$	4
					$\bar{PV}>TH_{PV}||SOC<SOC_{min}$	1

PT, Peak tariff period: regarding electricity, the most expensive period of the day. OT, Off-peak period: the most economical period of the day. $SOC_{min}$, minimum selected state of charge: this value is set according to the manufacturer’s guidelines for the BESS to maximize the relation between the depth of discharge of the BESS and the number of life cycles. $SOC_{max}$, maximum selected state of charge: this value is set according to the manufacture guidelines for the BESS to maximize the relation between the depth of discharge of the BESS and the number of life cycles. $TH_{PV}$, PV-selected threshold: this value sets the PV power required to switch to island mode ensuring that the BESS complements the PV with adequate discharge currents. $TH_{PV}^{-}$, PV-selected threshold minus offset: this value is set to establish a hysteresis in state transitions associated with the PV threshold, and thus, prevents high frequency transitions. $TH_{PV}^{+}$, PV-selected threshold plus offset: this value is set to establish a hysteresis in state transitions associated with the PV threshold, and thus, prevents high frequency transitions. $\bar{PV}$, average PV over the last 15 min: this is used to determine the continuity of the PV drop.

**Table 2 sensors-21-01202-t002:** Results summary.

	Characteristics	Annual Savings	Total Annual Savings
PV	Peak power = 3 kW	153.93 €/kW	461.78 €
BESS	Capacity = 9.12 kWh	16.8 €/kWh	153.26 €
	DoD = 60%		
	Efficiency = 85%		
	Cycle life = 1500		

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
