# Peer review of "Turning Base Transceiver Stations into Scalable and Controllable DC Microgrids Based on a Smart Sensing Strategy"

_sensors, 2021, doi:10.3390/s21041202_

Round 1

Reviewer 1 Report

The proposed concept is interesting. Here are my comments:

1) In this article, most of the idea is around controllable DC Microgrids and the corresponding energy management system. But the innovation point of the "smart sensing strategy" does not elaborate too much.

2) In the article, sections 1 to 3 needs to be improved in the structure and presentation. Some of them are overlapping in content. Especially, the introduction and review section needs to be shorter and concentrate. 

3) In Fig. 2, two dc/dc is existing, where one is located in the power system and another one is connected to the PV. What is the purpose of having two. Also, the load connection line is also considered as DC bus. It should not require a separate indicator.

4) The content in Fig. 2 and 3 are not matching with each other.

5) In Section 5.4, a program flow chat or logic diagram is recommended to add in order to let the audience understand the proposed logic clearly.

6) In this article, it does not mention how to determine the correct battery size. Are they the same as the traditional approach? Or what is the optimized size of the battery can be?  

Reviewer 2 Report

My comments are as follows:

  1. Paper length is a concern. The first three sections can be combined into "Introduction" and tightened up a bit.
  2. Figure 1 - spelling of antenna
  3. Line 291 - typing error
  4. For all the thousands of words, the authors have not described results very well. Figure 6 - please explain each plot and what it means in relation to the other plots in the figure.
  5. Figure 7 - some components are not labeled
  6. Please use only one style of presenting plots. Compare Figure 6 and Figure 10 - two different styles - avoid this.
  7. Table 1 is impossible to read. The copyeditor is not going to like the slovenly presentation. Please reconsider. If not essential, put it in the appendix as several smaller tables.
  8. Again, Tables 1 and 2 look like they came from two different papers. Uniformity is key.
  9. I really think Sections 4-5 contain an excessive amount of detail. No one is going to replicate this system precisely the way you have presented it. 75% of the paper need not be dedicated to this. The application and results and lessons learned are of far greater value.
  10. Figure 15 contains a wealth of information, but it dismissively mentioned with one sentence. Is the reader supposed to imagine what its significance is? Or would you please expand a bit? Are these results expected? Why or why not? What is the reason for the dips/spikes?
  11. Section 7 - ...and future work - not works and no period necessary (be consistent)

Reviewer 3 Report

The paper presents a solution to transform the electrical part of Base Transceiver Stations into electrical microgrid. The authors started from an analysis of uncertainty regarding the cost of operating a Base Transceiver Stations, but also from an analysis of the cost of various solutions for implementing a microgrid such as the use of renewable energy. The work continued with a state of the art on the solutions that are suitable for the realization of a microgrid suitable for Base Transceiver Stations.

The practical part of the paper consists of two chapters which describe in detail the architecture and construction of the microgrid from all points of view (electrical, hardware, software, management). The design and use of each hardware component is also presented in detail. The communication strategy between the various components of the microgrid is also well presented. Even the realization of a wireless weather station for the PV production forecast represents an important achievement in this work.

Chapter 6 presents the results obtained in the field, detailing the behavior of the microgrid during a day. At the end of the chapter there is also an economic study regarding the return on investment

The work is well organized with introduction, theoretical part, application part and conclusions. Each part is corrected presented. The case study provides enough information to understand the proposed solution and its cost-effectiveness. Conclusions are in correlation with the rest of the paper.

The work is of a high quality and I have no recommendation for its improvement

Round 2

Reviewer 1 Report

Nice work and it comes with a clear presentation. Here are my comments:

- a summary table may be a good approach to highlight the system specification and the  key finding to audience. 

- in the last sentence of 4.4, “ The total savings for the installed battery power is 153.26 euros/year without the correction and 676 euros/year applying the factor.” It may need to highlight the battery rating.

Author Response

The two suggestions of the reviewer have been addressed. Starting in line 677 this new text has been introduced:

Finally, Table 2 provides a summary of the obtained results for a BTS with 2kW of critical loads and 1 kW of non-critical load with energy supplied by the Spanish energy market. Considering a daily price differential between the peak and off-peak hours of approximately 0.09 €/kWh.

Table 2: Results Summary.

Characteristics

Annual savings

Total annual savings

PV

Peak power =3kW

153.93 €/kW

461.78 €

BESS

Capacity=9.12 kWh

DoD=60%

Efficiency=85%

Cycle life=1500

16.8 €/kWh

153.26 €

Reviewer 2 Report

In all my years of review, I can confidently say that this group of authors has made the most significant changes to their manuscript, which obviously required a huge effort. My comments were numerous and detailed, and the authors addressed each one of them winsomely and thoroughly. In my humble opinion, this revised version of the manuscript is an improvement over the previous version (which was already very good) and can be accepted for publication (I leave the treatment of the table in the appendix to the copyeditor). I am grateful to the authors for their effort and professional attitude. Best wishes going forward.

Author Response

Thank you for your comments.